# Diagnostic performance of serum interferon gamma, matrix metalloproteinases, and periostin measurements for pulmonary tuberculosis in Japanese patients with pneumonia

**Momoko Yamauchi[1], Takeshi Kinjo[1]\*, Gretchen Parrott[1], Kazuya Miyagi[1], Shusaku Haranaga[1,2], Yuko Nakayama[3], Kenji Chibana[3], Kaori Fujita[3], Atsushi Nakamoto[3], Futoshi Higa[3], Isoko Owan[3], Koji Yonemoto[4,5], Jiro Fujita[1]**

1 Department of Infectious, Respiratory, and Digestive Medicine, Graduate School of Medicine, University of the Ryukyus, Okinawa, Japan, 2 Center for General Clinical Training and Education, University of the Ryukyus Hospital, Okinawa, Japan, 3 Department of Respiratory Medicine, National Hospital Organization Okinawa Hospital, Okinawa, Japan, 4 Division of Biostatistics, School of Health Sciences, Faculty of Medicine, University of the Ryukyus, Okinawa, Japan, 5 Division of Biostatistics, Advanced Medical Research Center, Faculty of Medicine, University of the Ryukyus, Okinawa, Japan

\* t_kinjo@med.u-ryukyu.ac.jp

**Data Availability Statement:** All relevant data are within the manuscript and its Supporting Information files.

## Abstract

Serum markers that differentiate between tuberculous and non-tuberculous pneumonia would be clinically useful. However, few serum markers have been investigated for their association with either disease. In this study, serum levels of interferon gamma (IFN-γ), matrix metalloproteinases 1 and 9 (MMP-1 and MMP-9, respectively), and periostin were compared between 40 pulmonary tuberculosis (PTB) and 28 non-tuberculous pneumonia (non-PTB) patients. Diagnostic performance was assessed by analysis of receiver-operating characteristic (ROC) curves and classification trees. Serum IFN-γ and MMP-1 levels were significantly higher and serum MMP-9 levels significantly lower in PTB than in non-PTB patients (p < 0.001, p = 0.002, p < 0.001, respectively). No significant difference was observed in serum periostin levels between groups. ROC curve analysis could not determine the appropriate cut-off value with high sensitivity and specificity; therefore, a classification tree method was applied. This method identified patients with limited infiltration into three groups with statistical significance (p = 0.01), and those with MMP-1 levels < 0.01 ng/mL and periostin levels ≥ 118.8 ng/mL included only non-PTB patients (95% confidence interval 0.0–41.0). Patients with extensive infiltration were also divided into three groups with statistical significance (p < 0.001), and those with MMP-9 levels < 3.009 ng/mL included only PTB patients (95% confidence interval 76.8–100.0). In conclusion, the novel classification tree developed using MMP-1, MMP-9, and periostin data distinguished PTB from non-PTB patients. Further studies are needed to validate our cut-off values and the overall clinical usefulness of these markers.

**Funding:** The authors received no specific funding for this work.

**Competing interests:** The authors have declared that no competing interests exist.

## Introduction

Tuberculosis (TB) remains a leading cause of death, even in the era of established anti-TB treatment. Africa and South-East Asia account for 85% of TB deaths globally [1], and South-East Asia is estimated to bear one-third of the global TB burden [1]. A recent systematic review [2] found that more than 10% of cases of community-acquired pneumonia in Asia was caused by *Mycobacterium tuberculosis*. Prompt diagnosis of pulmonary tuberculosis (PTB) as the cause of pneumonia may be delayed because the early clinical presentations of PTB and non-tuberculous pneumonia (non-PTB) are often indistinguishable [3, 4].

Nucleic acid amplification is widely used to diagnose PTB but is not practical in patients with inadequate sputum quality and those who cannot expectorate. The interferon gamma (IFN-γ) release assay is also used to diagnose TB but has drawbacks, including the time required for a patient's lymphocytes to release IFN-γ against TB antigens in vitro and inability to differentiate active from latent TB infection (LTBI). Routine blood tests performed in patients with pneumonia could be helpful if a diagnostic marker in serum that distinguishes PTB from non-PTB is found.

IFN-γ is a type 2 interferon that plays a critical role in the immune response to TB infection by activating macrophages and other immune cells [5, 6]. Furthermore, some matrix metallo-proteinases (MMPs) are responsible for turnover, degradation, and catabolism of the extracel-lular matrix [7, 8]. MMP-1 and MMP-9 are reportedly associated with formation and cavitation of TB granuloma [9, 10]. Although IFN-γ, MMP-1, and MMP-9 levels in blood col-lected from patients with PTB have been demonstrated to be elevated [11–13], most studies have compared patients with PTB against healthy subjects or patients with LTBI. From the per-spective of clinical practice, it is important to investigate serum markers that can differentiate PTB from non-PTB, not from healthy subjects or LTBI patients.

Koguchi et al. reported that osteopontin, a matricellular protein, was elevated in patients with PTB [14]. Periostin, a similar matricellular protein, was shown to be a biomarker of chronic inflammation and fibrosis via its association with type 2 helper T-cell reactions in asthma and idiopathic pulmonary fibrosis [15–17]. Periostin is produced by fibroblasts and alveolar epithelial cells, and fibroblasts are known to have a role in formation of TB granuloma [18, 19]; therefore, periostin may also be elevated in patients with PTB. However, thus far, no studies have measured any of the aforementioned serum marker levels to discriminate between PTB and non-PTB.

The aims of this study were to investigate serum IFN-γ, MMP-1, MMP-9, and periostin lev-els in patients with PTB and those with non-PTB and to assess their value as diagnostic markers.

## Materials and methods

### Patients

Patients admitted to the University of the Ryukyus Hospital or the National Hospital Organi-zation Okinawa Hospital between January 2012 and December 2016 with a diagnosis of PTB or non-PTB in whom blood samples were collected before treatment were eligible for enrol-ment in the study. PTB was confirmed by a positive *M. tuberculosis* polymerase chain reaction test. The data collected included age, sex, underlying diseases, identification of causative bacte-ria, radiological findings, and serum IFN-γ, MMP-1, MMP-9, and periostin measurements. Serum samples collected between January 2012 and December 2016 were stored at -80°C until testing (between November 2016 and January 2018). The authors of the study had direct access to anonymized patient information. Patients were given the opportunity for opt-out via

websites of both the Department of Infectious, Respiratory, and Digestive Medicine, Graduate School of Medicine, University of the Ryukyus, and the National Hospital Organization Okinawa Hospital. Patients who wished not to participate in this study could be excluded via posted phone number on these websites. This study was approved by the Institutional Ethics Committees of both the University of the Ryukyus (approval number 1128) and the National Hospital Organization Okinawa Hospital (approval number 27–31).

### Radiographic assessment

Chest radiographs acquired at admission were scored from 0 to 10 based on the extent of infiltration as in previous reports [20, 21]. Scores of 0–4 indicated limited infiltration (in less than one-third of one lung; approximately 50% of patients) and scores of 5–10 indicated extensive infiltration (in one-third or more of one lung; approximately 50%).

### Assays

Serum IFN-γ levels were measured using the BD™ Cytometric Bead Array Human Th1/Th2/Th17 cytokine kit (Becton, Dickinson and Company, Franklin Lakes, NJ) via the BD Accuri C6 flow cytometer with a sequential multi-channel analyzer (Becton Dickinson). Serum periostin was measured using an enzyme-linked immunoassay kit (Phoenix Pharmaceuticals, Burlingame, CA). MMP-1 was measured using the Fluorokine E kit (R&D Systems, Minneapolis, MN) and MMP-9 using the Quantikine enzyme-linked immunoassay kit (R&D Systems). All assays were performed according to the manufacturers' instructions.

### Statistical analysis

The data for PTB and non-PTB patients are shown as the median (range) or number (percentage). The Wilcoxon rank-sum test was used to compare the continuous variables and the Pearson's chi-square or Fisher's exact test to compare the nominal variables.

Receiver-operating characteristic (ROC) curves were used to evaluate the diagnostic performance of each biomarker and to determine the specificity corresponding to high sensitivity (90%). A classification tree was also used. This method allows subjects to be sub-grouped by specific levels of explanatory variables and is developed by selecting explanatory variables with the smallest error classification rate of qualitative objective variables within groups [22]. In this study, the patients with extensive or limited infiltration were subgrouped using the classification tree with diagnosis (PTB or non-PTB) as the objective variable and IFN-γ, MMP-1, MMP-9, and periostin as the explanatory variables. The 95% confidence interval (CI) for prevalence of PTB in each subgroup was calculated using the Clopper-Pearson method. All statistical analyses were performed using JMP version 14 (SAS Institute Inc., Cary, NC). A two-sided p-value < 0.05 was considered statistically significant.

## Results

Fifty-eight of 92 patients with pneumonia had a diagnosis of PTB and 34 did not. Eighteen patients in the PTB group were excluded because of a negative TB-polymerase chain reaction result (n = 7) or having no pre-treatment blood sample (n = 11) and 6 in the non-PTB group were excluded for having no pre-treatment blood sample available (n = 3) or an inadequate sample volume (n = 3). After these exclusions, 40 patients were enrolled in the PTB group and 28 in the non-PTB group (Fig 1). The demographics, underlying diseases, and radiographic findings of pneumonia in the study groups are summarized in Table 1. The median age was 78 (range 46–100) years in the PTB group and 74 (range 55–91) years in the non-PTB group.

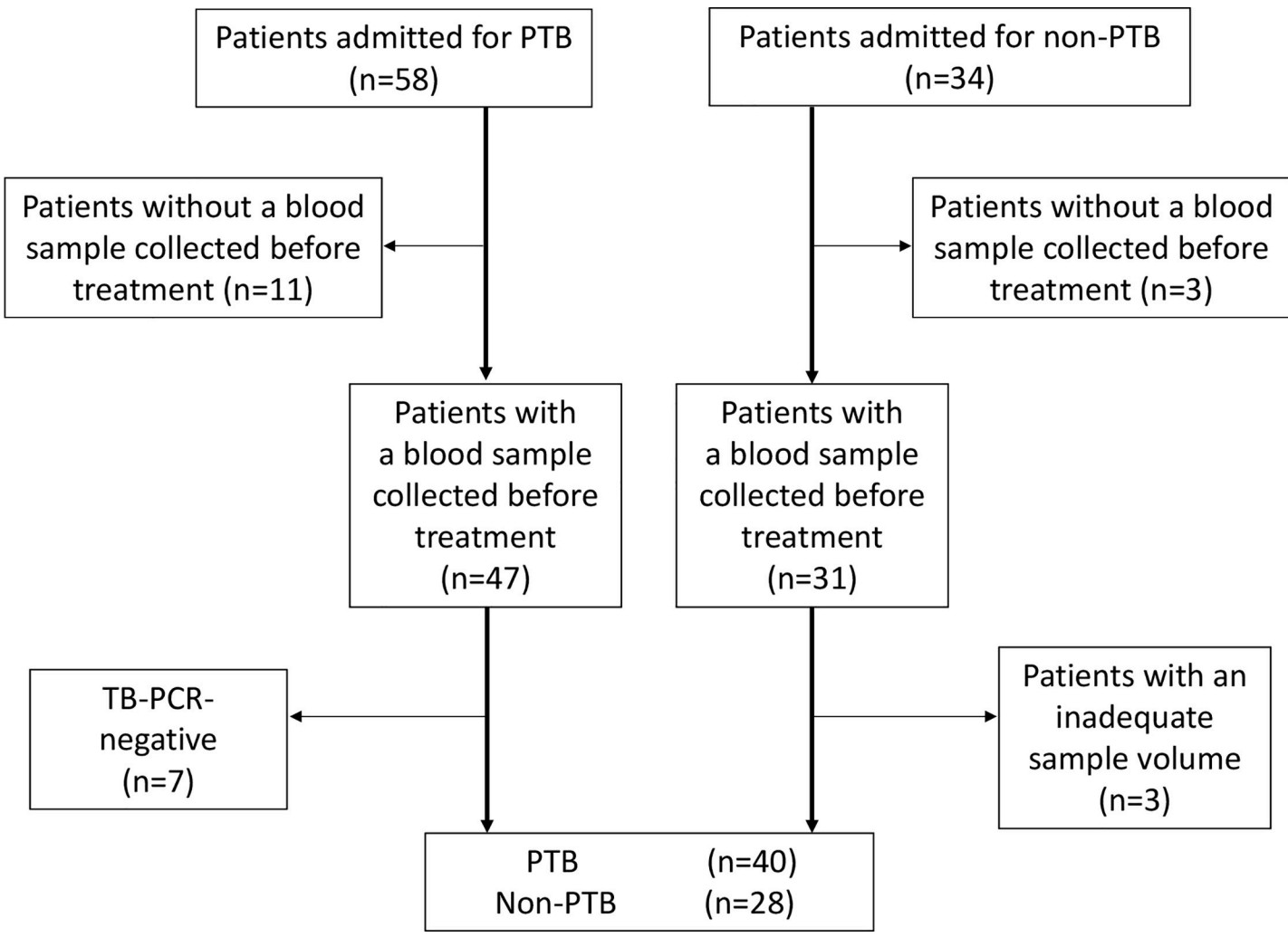

**Fig 1. Flow chart showing the selection of PTB and non-PTB patients in this study.** Eligible patients were further extracted by the inclusion criteria described in the Methods section. PCR, polymerase chain reaction; PTB, pulmonary tuberculosis.

There were no significant between-group differences in sex distribution or the frequency of underlying diseases. Bilateral lesions were present in 65.0% of patients in the PTB group and 57.1% of those in the non-PTB group (p = 0.51) and pleural effusions were present in 22.5% and 17.9%, respectively (p = 0.64); extensive infiltrations were seen on chest radiographs in 72.5% and 39.3% (p = 0.006) and cavities in 32.5% and 0% (p < 0.001). The leading causative pathogen in the non-PTB group was *Haemophilus influenzae* (25.0%) followed by *Streptococcus pneumoniae* (17.9%) and *Klebsiella pneumoniae* (10.7%; S1 Table).

Serum IFN-γ and MMP-1 levels were significantly higher in the PTB group (p < 0.001 and p = 0.002, respectively) whereas MMP-9 levels were significantly lower (p < 0.001; Fig 2). There was no significant between-group difference in the periostin level. The median (interquartile range) IFN-γ, MMP-1, MMP-9, and periostin levels in the PTB group were 3.46 pg/dL (0.11–10.11), 0.024 ng/dL (0.00–0.082), 3.45 ng/dL (1.83–7.81), and 198.86 ng/dL (105.23–289.57), respectively; the respective values in the non-PTB group were 0.00 pg/dL (0.00–0.28), 0.0 ng/dL (0.00–0.0088), 8.66 ng/dL (4.58–12.04), and 131.06 ng/dL (74.80–166.58). Because factors other than PTB may affect serum IFN-γ and MMP-1 levels, we compared these

**Table 1. Patient demographics, underlying diseases, and radiographic findings of pneumonia.**

| | PTB | Non-PTB | |
|---|---|---|---|
| | (n = 40) | (n = 28) | p-value[*] |
| Age | 78 (46–100) | 74 (55–91) | 0.28 |
| Female sex[#] | 40.0% (16) | 17.9% (5) | 0.052 |
| Underlying disease | | | |
| ILD | 5.0% (2) | 7.1% (2) | >0.99 |
| Asthma | 2.5% (1) | 7.1% (2) | 0.56 |
| COPD | 7.5% (3) | 21.4% (6) | 0.15 |
| Bronchiectasis | 2.5% (1) | 3.6% (1) | >0.99 |
| Malignancy | 12.5% (5) | 7.1% (2) | 0.69 |
| Cardiovascular[#] | 22.5% (9) | 28.6% (8) | 0.57 |
| CKD (non-HD) | 7.5% (3) | 14.3% (4) | 0.43 |
| Chronic liver disease | 2.5% (1) | 3.6% (1) | >0.99 |
| Hemodialysis | 7.5% (3) | 7.1% (2) | >0.99 |
| Diabetes mellitus[#] | 27.5% (11) | 14.3% (4) | 0.20 |
| HIV infection | 0.0% (0) | 7.1% (2) | 0.17 |
| Immunosuppressant | 2.5% (1) | 0.0% (0) | >0.99 |
| Radiographic findings of pneumonia[#] | | | |
| Bilateral disease | 65.0% (26) | 57.1% (16) | 0.51 |
| Extensive infiltration | 72.5% (29) | 39.3% (11) | 0.006 |
| Pleural effusion | 22.5% (9) | 17.9% (5) | 0.64 |
| Cavity | 32.5% (13) | 0.0% (0) | < 0.001 |

The median age (range) and proportion (number) of female patients, patients with each underlying disease, immunosuppressant use, and patients with each radiographic finding are shown.

[*]Differences between PTB and non-PTB were analyzed by Fisher's exact test, except variables labelled with the hash sign.

[#]Differences between two groups were analyzed using the Pearson's chi-square test. Abbreviations: PTB, pulmonary tuberculosis; ILD, interstitial lung disease; COPD, chronic obstructive pulmonary disease; CKD, chronic kidney disease; HD, hemodialysis; HIV, human immunodeficiency virus

biomarkers by gender and underlying diseases regardless of the presence of PTB. The results show that there were no significant differences in serum IFN-γ levels by gender or underlying diseases. In contrast, serum MMP-1 levels in patients with malignancy were significantly higher than those without malignancy (S1 Fig). However, multivariate analysis showed that only PTB was associated with elevated serum MMP-1 levels (S2 Table).

The diagnostic performances of IFN-γ, MMP-1, MMP-9, and periostin were assessed using the ROC curves. The respective areas under the curve for IFN-γ, MMP-1, MMP-9, and periostin were 0.71 (95% CI 0.49–0.87), 0.69 (95% CI 0.55–0.94), 0.75 (95% CI 0.53–0.89), and 0.45 (95% CI 0.24–0.69) in the patients with limited infiltration and 0.79 (95% CI 0.56–0.91), 0.71 (95% CI 0.65–0.91), 0.81 (95% CI 0.63–0.92), and 0.63 (95% CI 0.42–0.81) in those with extensive infiltration (Fig 3). However, when a cut-off value with high sensitivity (90%) was used, the respective specificity values for IFN-γ, MMP-1, MMP-9, and periostin were as low as 0.24, 0.18, 0.47, and 0.00 in the limited infiltration group and 0.35, 0.16, 0.46, and 0.27 in the extensive infiltration group.

The diagnostic performance of each of the four proteins was then evaluated using the classification tree (Fig 4). Six of the patients with limited infiltration in the PTB group and two in the non-PTB group had an MMP-1 level ≥ 0.01 ng/mL (Fig 4, box A); none of those with an MMP-1 level < 0.01 ng/mL in the PTB group had a periostin level ≥118.8 ng/mL (Fig 4, box C). Five patients in the PTB group and eight in the non-PTB group had an MMP-1

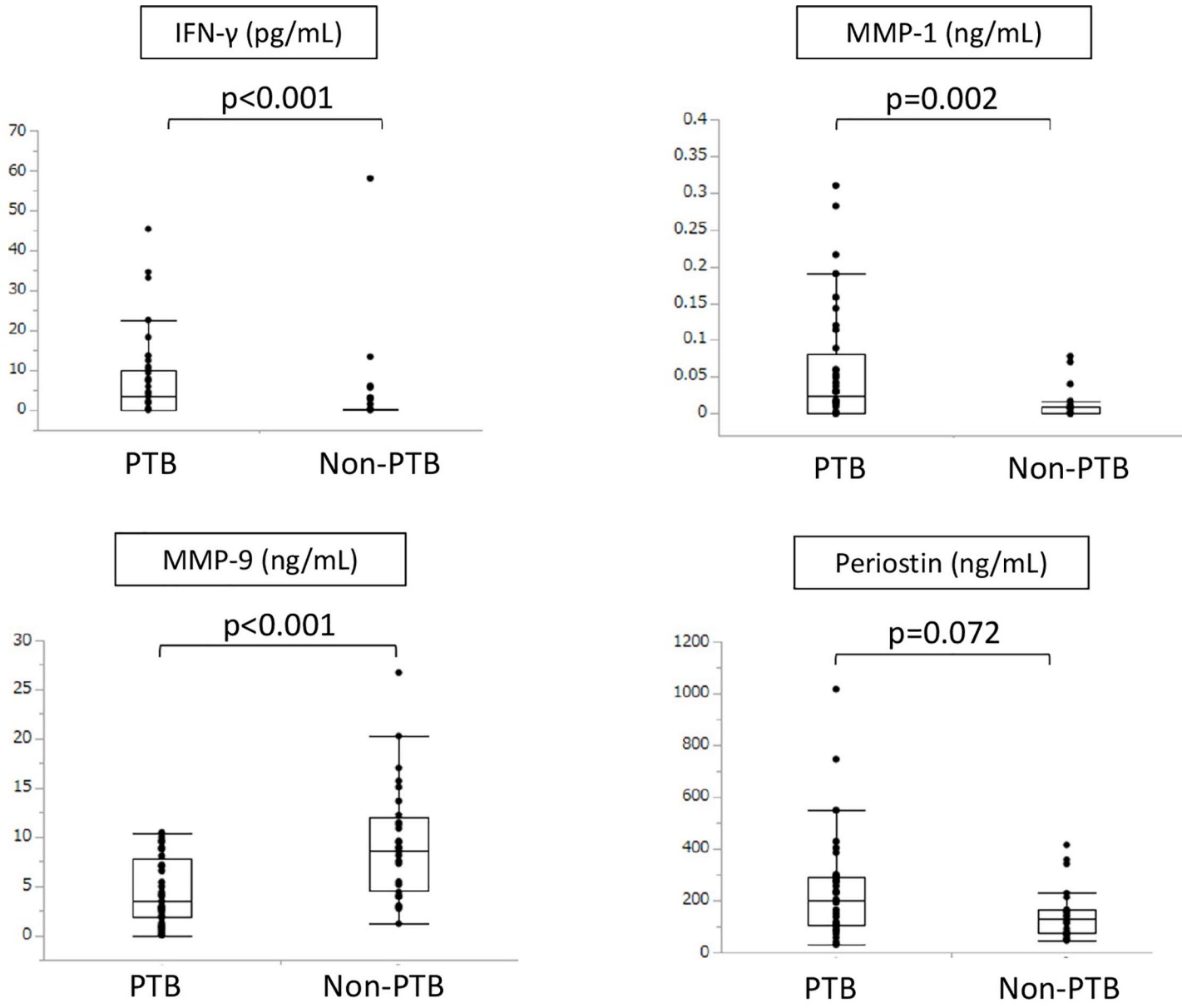

**Fig 2. Comparison of serum markers between PTB and non-PTB patients.** Serum IFN-γ, MMP-1, MMP-9, and periostin levels were compared between the two groups of patients. IFN-γ, interferon gamma; MMP-1, matrix metalloprotein-1; MMP-9, matrix metalloprotein-9.

level < 0.01 ng/mL and a periostin level < 118.8 ng/mL (Fig 4, box B); the respective values in the PTB group were 75% (95% CI 34.9–96.8), 38.5% (95% CI 13.9–68.4), and 0% (95% CI 0.0–41.0; Fig 5A). Significant differences were found between the three subgroups (p = 0.01). All patients with extensive infiltration and an MMP-9 level < 3.009 ng/mL were in the PTB group (Fig 4, box D). Fifteen of the patients with an MMP-9 level in the range of 3.009–10.844 ng/mL were in the PTB group and 6 were in the non-PTB group. All patients with an MMP-9 level ≥ 10.844 ng/mL were in the non-PTB group. The proportions of patients with PTB in boxes D, E, and F were 100% (95% CI 76.8–100), 71.4% (95% CI 47.8–88.7), and 0% (95% CI 0.0–52.2), respectively. The between-group differences were statistically significant (p < 0.001, Fig 5B).

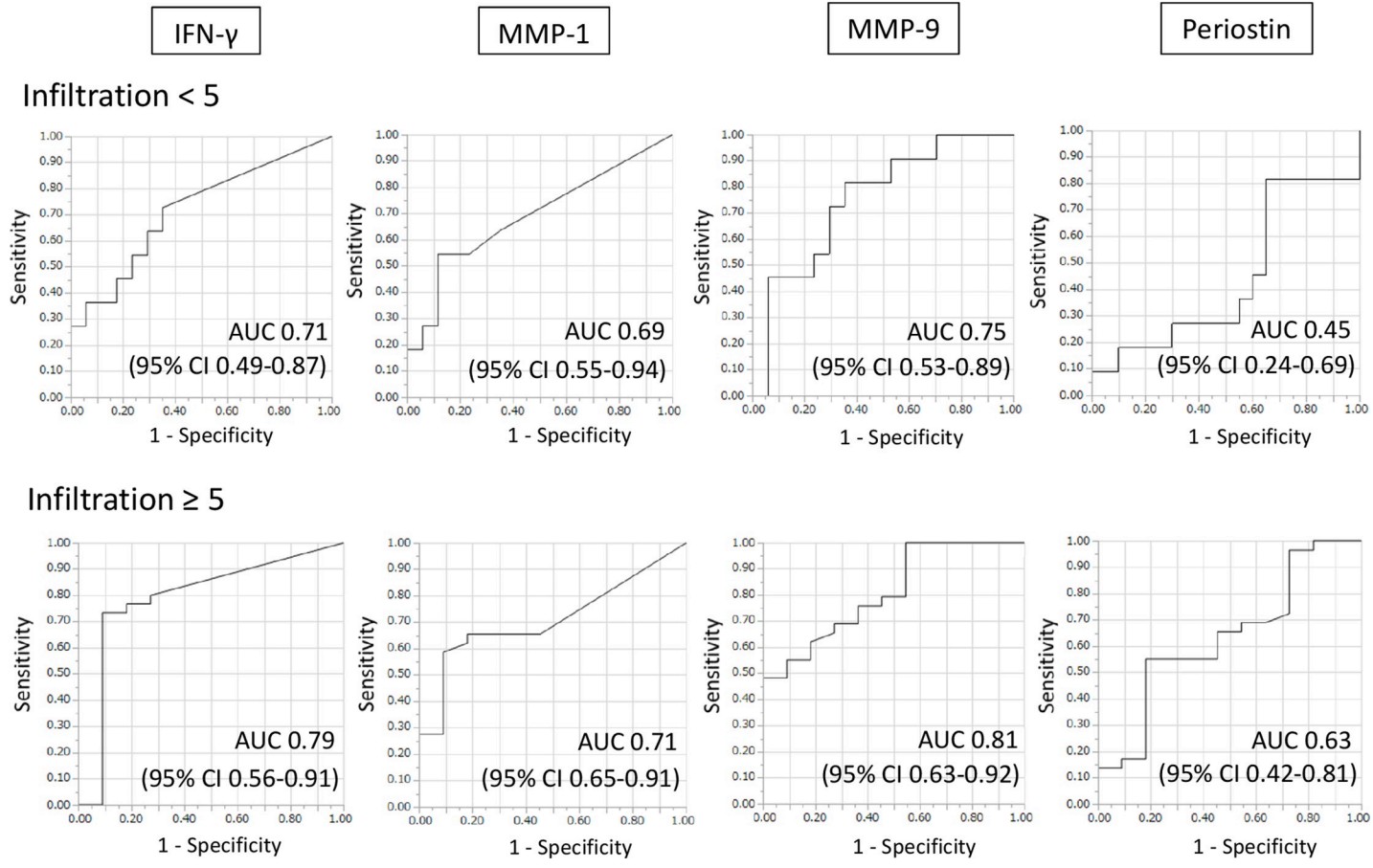

**Fig 3. Diagnostic performance in PTB patients with limited infiltration and their counterparts with extensive infiltration.** ROC curves showing the diagnostic performance of IFN-γ, MMP-1, MMP-9, and periostin for PTB in patients with radiographic evidence of limited or extensive infiltration. AUC, area under the curve; CI, confidence interval; IFN-γ, interferon gamma; MMP-1, matrix metalloprotein-1; MMP-9, matrix metalloprotein-9.

## Discussion

This study provides early evidence of the ability of IFN-γ, MMP-1, MMP-9, and periostin to differentiate between PTB and non-PTB patients. Although IFN-γ, MMP-1, and MMP-9 are known to be crucial players in the immune response to TB, few studies have investigated their value as markers for differentiating between PTB and non-PTB. Moreover, there are no reports on serum periostin levels in patients with PTB.

Previous reports have demonstrated elevated serum IFN-γ levels in patients with PTB when compared with healthy controls [11, 23, 24]. Yamada et al. reported that the IFN-γ level was significantly higher in patients with radiographic evidence of far-advanced PTB than in their counterparts with minimal or moderately advanced PTB and in healthy controls [11]. In the present study, there was no obvious association between the IFN-γ level and the extent of infiltration in patients with PTB (data not shown), possibly because of individual variation in the immune response and the fact that patients with malignancy, collagen disease, and immunosuppressive therapy were included, unlike in the study by Yamada et al. Another possible reason for this discrepancy may be the time interval between the onset of symptoms and blood collection; however, this interval was not recorded in either study. Min et al. reported that the serum IFN-γ level reached a peak in rhesus monkeys at 6 weeks after infection with *M.*

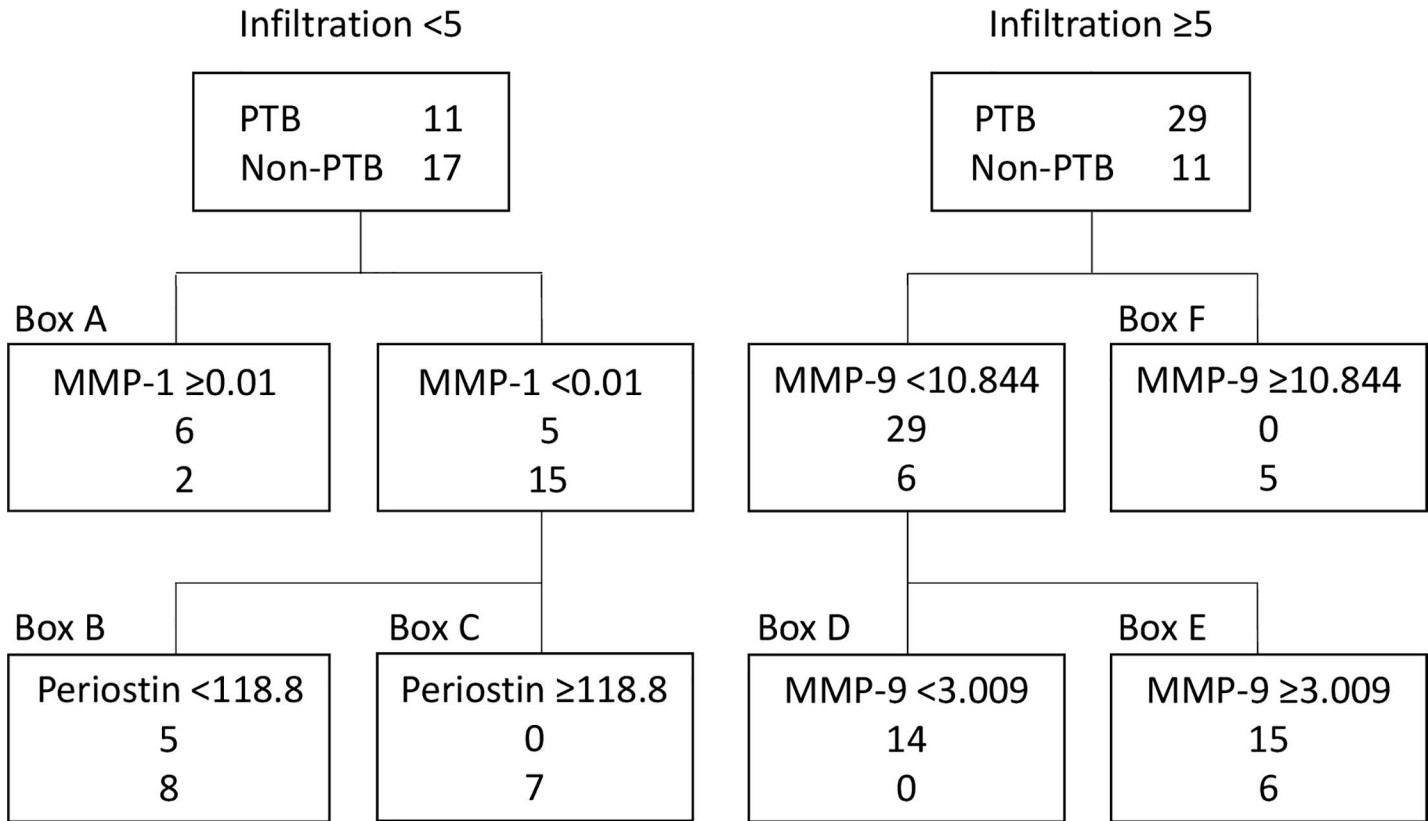

**Fig 4. Algorithm for discrimination between PTB and non-PTB patients using the classification tree method.** Serum markers were used to differentiate between PTB and non-PTB patients according to the extent of infiltration using the classification tree. The cut-off values used to discriminate between the patient groups are shown on the uppermost line. The second number shows the number of patients with PTB. The lowest line shows the number of non-PTB patients affected by the cut-off value. IFN-γ, interferon gamma; MMP-1, matrix metalloprotein-1; MMP-9, matrix metalloprotein-9; PTB, pulmonary tuberculosis.

*tuberculosis* and returned to baseline after 12 weeks [25]. Therefore, timing of blood collection may influence the IFN-γ level.

MMP-1 can degrade the extracellular (collagen) matrix and destroy caseous granuloma, leading to formation of cavities in the lungs of patients with PTB, and is released during cavity formation in a tuberculous lesion regardless of the existence of cavities [9, 26, 27]. Previous reports have demonstrated plasma MMP-1 levels to be significantly higher in patients with PTB than in healthy controls and patients with LTBI or sarcoidosis [12, 28]. To date, no reports have compared serum MMP-1 levels between PTB and non-PTB patients. Our results show that serum MMP-1 levels in PTB were still higher than those in non-PTB, suggesting MMP-1 can be used to distinguish PTB from non-PTB patients.

We found that the serum MMP-9 levels were significantly lower in PTB than in non-PTB patients, whereas Xu et al. reported that serum MMP-9 levels were significantly higher in PTB than in non-PTB patients [13]. However, there are several reports of increased serum MMP-9 in patients with community-acquired or ventilator-associated pneumonia [29–31]. In contrast, Hrabec et al. reported that the serum MMP-9 level was three times higher in patients with TB than in normal subjects [32]. Therefore, MMP-9 may be increased in both PTB and non-PTB. Our data suggest that the MMP-9 level may be lower in patients with PTB; however, further studies are needed to explain the discrepancy between our finding and that of Xu et al.

ROC curves were created to investigate the diagnostic performance of each protein for PTB. Given that the extent of infiltration is likely to be involved, the patients were divided

## (a) Proportion of patients with PTB and limited infiltration

|  | Proportion for PTB (%) | 95% CI* (%) |
|---|---|---|
| Box A | 75.0 | 34.9-96.8 |
| Box B | 38.5 | 13.9-68.4 |
| Box C | 0.0 | 0.0-41.0 |

## (b) Proportion of patients with PTB and extensive infiltration

|  | Proportion with PTB (%) | 95% CI* (%) |
|---|---|---|
| Box D | 100.0 | 76.8-100.0 |
| Box E | 71.4 | 47.8-88.7 |
| Box F | 0.0 | 0.0-52.2 |

**Fig 5. Proportion of PTB patients with limited or extensive infiltration.** The proportions and 95% confidence intervals for PTB patients with limited infiltration (a) and extensive infiltration (b) are shown. *Confidence interval. PTB, pulmonary tuberculosis.

according to whether the infiltration was limited or extensive. MMP-9 was the most useful of the four proteins for discriminating PTB from non-PTB regardless of the extent of infiltration. However, when a protein cut-off value with high sensitivity (90%) was determined, the corresponding specificity was low, so we considered it inappropriate to using these cut-off values to differentiate between PTB and non-PTB. Therefore, a classification tree was created to evaluate the clinical value of each marker. MMP-1 and periostin were selected as the explanatory variables in the group with limited infiltration and MMP-9 as the explanatory variable in the group with extensive infiltration. In the patients with limited infiltration, the 95% CI for the proportion with PTB in box C in Fig 4 was 0–41, indicating that these patients were likely not to have PTB. Moreover, the 95% CI for the patients with extensive infiltration in box D was 76.8–100, implying a high likelihood of PTB. The 95% CIs were broad in the rest of the boxes, so the likelihood of those patients having PTB is unclear. Nevertheless, this classification could be useful in the clinical management of patients, especially those presenting with pneumonia

of unknown origin. The classification tree method automatically chooses some appropriate markers and their cut-off values to divide patients into two groups according to its algorithm [22]. Although interpretation of the classification is sometimes difficult, it is possible that MMP-9 was chosen as a useful biomarker especially in patients with extensive infiltration because it was reported that plasma MMP-9 levels are associated with severity of ventilator-associated pneumonia [29]. Therefore, the difference between PTB and non-PTB may become more apparent in patients with extensive infiltration.

Our study has several strengths. First, it compared IFN-γ, MMP-1, MMP-9, and periostin levels between patients with PTB and non-PTB. Except for one study by Xu et al. [13], who focused on MMP-9, previous studies compared these proteins between patients with PTB and healthy controls or patients with LTBI. Given the clinical importance of identifying PTB in patients with pneumonia, our study design is more appropriate for identification of clinically useful diagnostic serum markers. Second, both ROC analysis and a classification tree were used to evaluate the clinical usefulness of these markers. Unlike ROC analysis, a classification tree could prioritize each of the markers and classify patients into subgroups based on their risk, so it was the appropriate method for identifying patients with a high probability of PTB.

There are also some limitations to this research. First, the sample size was small, so the 95% CIs were broad. However, exclusion of patients without pre-treatment blood samples ensured the validity of the study. Second, the study had a retrospective design and some patients were excluded because of our selection criteria, which may have introduced a degree of selection bias. This possibility is considered small because no patients were excluded according to their characteristics. Finally, the study included data from only two hospitals. However, in Okinawa, most patients with PTB who require admission are referred to one of these two institutions, so our results can be considered representative for this region.

In conclusion, serum IFN-γ, MMP-1, MMP-9, and periostin levels could be useful markers for distinguishing between PTB and non-PTB. However, our findings need to be confirmed in larger studies that include more centers before they can be generalized to other populations.

## Supporting information

**S1 Table. Causative bacteria in patients with non-tuberculous pneumonia.**
(DOC)

**S2 Table. Factors associated with elevated MMP-1 levels by multivariate analysis.**
(DOC)

**S1 Fig. Comparison of serum MMP-1 levels between patients with and without malignancy.**
(TIFF)

**S1 File. Raw data.**
(XLSX)

## Acknowledgments

We would like to thank laboratory technicians in the Department of Infectious Diseases, Respiratory, and Digestive Medicine, Graduate School of Medicine, University of the Ryukyus, and National Hospital Organization Okinawa Hospital, for their contribution to handling serum samples. We would also like to thank *Editage* (www.editage.com) for English language editing.

## Author Contributions

**Conceptualization:** Momoko Yamauchi, Takeshi Kinjo, Koji Yonemoto, Jiro Fujita.

**Data curation:** Momoko Yamauchi, Takeshi Kinjo, Gretchen Parrott, Kazuya Miyagi, Shu-saku Haranaga, Yuko Nakayama, Kenji Chibana, Kaori Fujita, Atsushi Nakamoto, Futoshi Higa, Isoko Owan.

**Formal analysis:** Momoko Yamauchi, Takeshi Kinjo, Gretchen Parrott, Koji Yonemoto.

**Investigation:** Momoko Yamauchi, Takeshi Kinjo, Jiro Fujita.

**Methodology:** Momoko Yamauchi, Takeshi Kinjo, Gretchen Parrott, Koji Yonemoto.

**Supervision:** Takeshi Kinjo, Koji Yonemoto, Jiro Fujita.

**Validation:** Momoko Yamauchi, Takeshi Kinjo, Gretchen Parrott, Koji Yonemoto.

**Writing – original draft:** Momoko Yamauchi, Takeshi Kinjo, Koji Yonemoto.

**Writing – review & editing:** Momoko Yamauchi, Takeshi Kinjo, Gretchen Parrott, Kazuya Miyagi, Shusaku Haranaga, Yuko Nakayama, Kenji Chibana, Kaori Fujita, Atsushi Nakamoto, Futoshi Higa, Isoko Owan, Koji Yonemoto, Jiro Fujita.

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
