## [Decision Letter · Decision Letter 0]

20 Aug 2019

PONE-D-19-19256

Diagnostic performance of serum interferon gamma, matrix metalloproteinases, and periostin measurements for pulmonary tuberculosis in Japanese patients with pneumonia

PLOS ONE

Dear Dr. Kinjo,

Thank you for submitting your manuscript to PLOS ONE. After careful consideration, we feel that it has merit but does not fully meet PLOS ONE’s publication criteria as it currently stands. Therefore, we invite you to submit a revised version of the manuscript that addresses the points raised during the review process.

We would appreciate receiving your revised manuscript. To enhance the reproducibility of your results, we recommend that if applicable you deposit your laboratory protocols in protocols.io, where a protocol can be assigned its own identifier (DOI) such that it can be cited independently in the future. For instructions see: http://journals.plos.org/plosone/s/submission-guidelines#loc-laboratory-protocols

We look forward to receiving your revised manuscript.

Kind regards,

Frederick Quinn

Academic Editor

PLOS ONE

Journal Requirements:

2. In the ethics statement in the manuscript and in the online submission form, please provide additional information about the patient records/samples used in your retrospective study. Specifically, please ensure that you have discussed whether all data/samples were fully anonymized before you accessed them and/or whether the IRB or ethics committee waived the requirement for explicit informed consent (an opt-out system does not necessarily fulfil this criterion). If patients provided informed written consent to have data/samples from their medical records used in research, please include this information.

3. Please note that all PLOS journals ask authors to adhere to our policies for sharing of data and materials: https://journals.plos.org/plosone/s/data-availability. According to PLOS ONE’s Data Availability policy, we require that the minimal dataset underlying results reported in the submission must be made immediately and freely available at the time of publication. As such, please remove any instances of 'unpublished data' or 'data not shown' in your manuscript and replace these with either the relevant data (in the form of additional figures, tables or descriptive text, as appropriate), a citation to where the data can be found, or remove altogether any statements supported by data not presented in the manuscript.

Reviewers' comments:

Reviewer's Responses to Questions

**Comments to the Author**

1. Is the manuscript technically sound, and do the data support the conclusions?

Reviewer #1: Partly

Reviewer #2: No

2. Has the statistical analysis been performed appropriately and rigorously? 

Reviewer #1: Yes

Reviewer #2: Yes

3. Have the authors made all data underlying the findings in their manuscript fully available?

Reviewer #1: Yes

Reviewer #2: Yes

4. Is the manuscript presented in an intelligible fashion and written in standard English?

Reviewer #1: No

Reviewer #2: No

5. Review Comments to the Author

Reviewer #1: Yamauchi et al assess the the Diagnostic performance of serum IFNg, MMP and periostin measurements for PTB in Japanese population. For this, they first performed ROC analysis and found IFNg, MMP-1 higher and MMP-9 lower in PTB pneumonia. The authors further adopt a classification tree based on cut-off values of MMP- 1, -9 and periostin to distinguish PTB and non-PTB pneumonia patients. Since the TB diagnostic field needs a biomarker that has better prognostic value than IFNg particularly in TB endemic regions, the proposed work is appealing as it explores periostin as a novel molecule in PTB. Periostin is an important protein in asthma and airway inflammation and is worth studying its role during M tuberculosis infection. However, this research has several drawbacks and I have reservations in accepting the manuscript in the present form.

One of my major concern is that, the manuscript is not well and thoughtfully written.

- The use of PTB pneumonia is confusing, do authors use this term in place of PTB or PTB pneumonia has clinical features distinct from PTB. It should be explained in the Introduction.

- The Results section and Figure Legends are mixed-up, which is confusing and breaks the continuity of reading the manuscript. Separating out the two sections would have been better.

- In the Discussion section line 240, it is not clear what authors mean by "....early evidence....", is it combination of the 4 biomarkers determine early disease stage or the study is the new evidence of ability of these biomarkers to determine PTB vs non-PTB? Line 260, authors start with description of MMP-1, state no reports of MMP-1 (line 265) and suddenly switch to their findings for MMP-9 (line 267), without discussing about their opinion on MMP-1.

- The study design has a Major flaw that is does not include age matched Healthy control population. Further, sample size is too small to reach any logical conclusion.

- What were the biomarker(s) levels pre-treatment? Does serum Periostin, MMP-1 and MMP-9 levels changed post anti-tuberculosis treatment?

- It is my understanding that the BD CBA Th1/Th2/Th17 used in this study can also determine IL-2,-6,-17A and TNFa quantities. Postn-/- mice increase TNFa production (Carcinogenesis 2019; 40 (1): 102); since TNFa is a prominent cytokine with anti-TB activity, it will be interesting and important to assess modulation of these cytokines in relation to periostin in PTB and non-PTB conditions.

- The claim made by authors to distinguish PTB and non-PTB patients using classification tree method does not corroborate with the results (Figure 4). Firstly, Figure Legend has IFNg included, but neither the Figure 4, nor the text has any mention of IFNg in this analysis. Line 211 explains, all patients (infiltration <5) with MMP-1 <0.01 ng/ml in the non-PTB group had periostin level >118.8 ng/ml, when in fact 7 (46.7%) had > 118.8 and 8 (53.3%) had <118.8 ng/ml periostin level. If periostin is not significantly different in PTB and non-PTB groups, what was the purpose to include it is classification tree and how is it helpful to discriminate the patient groups?

- Similarly, Infiltration>5 group, nearly equal numbers of PTB patients have MMP-9 <3.009 (N=14) and >3.009 (N=15).What is the relevance of these values.

- I highly recommend along with Figure 5 (This is actually a Table) authors include a summary diagram (diagnostic algorithm) that is easy to follow and explains criteria / variables / cut-off values proposed by the authors to discriminate PTB and non-PTB groups.

Reviewer #2: In this study, the authors analyzed the serum levels of IFN-γ, MMP-1, MMP-9 and periostin in tuberculous pneumonia and non-tuberculous pneumonia patients. They also developed a new classification tree by using MMP-1, MMP-9 and periostin level to diagnose tuberculous pneumonia and non-tuberculous pneumonia patients. Their results are of interest. However, there are several major concerns that need to be addressed.

1. MMP-1 and MMP-9 are well known factors involved in many diseases. They are not specific markers for tuberculosis. The sample size in this study is small (40 vs. 28) and patients in each group are with a wide range of ages and diseases. I think it will be more appropriate to perform a more specific analysis. E.g. compare PTD vs non-PTD when both group patients have cardiovascular disorder or bilateral disease (with similar condition)…

2. The authors need to clarify the expression profile of MMP-1, MMP-9 and periostin over the infection time . Is there any dynamics? I am not clear if they compared the samples from the same time after infection.

3. I hope they can provide more explanation on the association between MMP-9 level and pneumonia patients with limited infiltration; also for the association between MMP-1, periostin and pneumonia patients with extensive infiltration.

4. I feel a little bit hard to read this manuscript due to the language. I suggest they have someone to correct the grammar and typos.

6. PLOS authors have the option to publish the peer review history of their article (what does this mean?). If published, this will include your full peer review and any attached files.

Reviewer #1: No

Reviewer #2: No

---

## [Author Response · Author response to Decision Letter 0]

9 Dec 2019

Reviewer #1: Yamauchi et al assess the the Diagnostic performance of serum IFNg, MMP and periostin measurements for PTB in Japanese population. For this, they first performed ROC analysis and found IFNg, MMP-1 higher and MMP-9 lower in PTB pneumonia. The authors further adopt a classification tree based on cut-off values of MMP- 1, -9 and periostin to distinguish PTB and non-PTB pneumonia patients. Since the TB diagnostic field needs a biomarker that has better prognostic value than IFNg particularly in TB endemic regions, the proposed work is appealing as it explores periostin as a novel molecule in PTB. Periostin is an important protein in asthma and airway inflammation and is worth studying its role during M tuberculosis infection. However, this research has several drawbacks and I have reservations in accepting the manuscript in the present form.

We really appreciate your reading of our manuscript and your comments and suggestions. We revised our manuscript following your comments. It would be most appreciated if you could give us your feedback to our response.

One of my major concern is that, the manuscript is not well and thoughtfully written.

- The use of PTB pneumonia is confusing, do authors use this term in place of PTB or PTB pneumonia has clinical features distinct from PTB. It should be explained in the Introduction.

Thank you for your comments. We totally agree that the submitted version of our manuscript was confusing owing to the wording you pointed out. In the revised version, we used “PTB” or “non-PTB” to make the text clearer.

- The Results section and Figure Legends are mixed-up, which is confusing and breaks the continuity of reading the manuscript. Separating out the two sections would have been better.

This “mixed-up” style is required by the submission guidelines of PLOS ONE journal.

- In the Discussion section line 240, it is not clear what authors mean by "....early evidence....", is it combination of the 4 biomarkers determine early disease stage or the study is the new evidence of ability of these biomarkers to determine PTB vs non-PTB? 

We used the expression “early evidence” to express that our data were not conclusive but showed the potential of the identified biomarkers to differentiate PTB from non-PTB. We think our findings should be validated by additional studies in the future as described in the Discussion section of the revised manuscript.

Line 260, authors start with description of MMP-1, state no reports of MMP-1 (line 265) and suddenly switch to their findings for MMP-9 (line 267), without discussing about their opinion on MMP-1.

Thank you for your comments. Following your suggestions, we added our opinion regarding the findings of MMP-1 (lines 270-272). In addition, the description of MMP-9 now starts in a new paragraph (Line 273).

- The study design has a Major flaw that is does not include age matched Healthy control population. Further, sample size is too small to reach any logical conclusion.

As we already mentioned in the Discussion section (lines 309-311), we think one of the strengths of our study is the comparison of serum biomarkers between PTB and non-PTB patients, not healthy subjects. From the perspective of clinical practice, physicians should differentiate PTB from non-PTB patients, not these from healthy subjects. Therefore, serum biomarkers that are higher in patients than in healthy subjects are not helpful to differentiate PTB from non-PTB in clinical practice. To make this point clearer, we added a description to the Introduction section (lines 72-74). In terms of the sample size in our study, we totally agree with your comment. As we mentioned in the Discussion section, this is a limitation of our study. However, we excluded patients without enough pre-treatment blood samples to ensure the validity of the study. Our findings are not conclusive, but we believe our work firstly showed the potential of these biomarkers as a diagnostic aid for PTB, thus it is worthy to report.

- What were the biomarker(s) levels pre-treatment? Does serum Periostin, MMP-1 and MMP-9 levels changed post anti-tuberculosis treatment?

Thank you for your comments. As we explained in the Materials and Methods section, all sera used in the study were collected before TB treatment. We were also interested in the kinetics of these biomarkers during the course of TB treatment. In fact, we compared serum periostin levels between pre- (N=16) and post-TB (N=16) treatment, although no significant difference was found. We totally agree that the data regarding kinetics of these biomarkers deepen our understanding. However, this was not the primary purpose of the study, and serum samples collected after TB treatment were stored in only a subset of patients. This issue will be clarified in future studies.

- It is my understanding that the BD CBA Th1/Th2/Th17 used in this study can also determine IL-2,-6,-17A and TNFa quantities. Postn-/- mice increase TNFa production (Carcinogenesis 2019; 40 (1): 102); since TNFa is a prominent cytokine with anti-TB activity, it will be interesting and important to assess modulation of these cytokines in relation to periostin in PTB and non-PTB conditions.

Thank you for your comments. We actually measured IL-2, IL-6, IL-17A, and TNF-alpha levels with the BD CBA Th1/Th2/Th17 kit, although there were no significant differences in the levels of these cytokines between the PTB and non-PTB groups. Because these cytokines were not our focus, we did not include the corresponding data in our manuscript. We also appreciate your comment regarding the relation between TNF-alpha and periostin. From the viewpoint of basic immunology, it may be important to further evaluate the association between these biomarkers. However, the purpose of this study was to investigate potential biomarkers to help PTB diagnosis in a clinical setting, and therefore we did not perform association analysis among biomarkers. 

- The claim made by authors to distinguish PTB and non-PTB patients using classification tree method does not corroborate with the results (Figure 4). Firstly, Figure Legend has IFNg included, but neither the Figure 4, nor the text has any mention of IFNg in this analysis. Line 211 explains, all patients (infiltration <5) with MMP-1 <0.01 ng/ml in the non-PTB group had periostin level >118.8 ng/ml, when in fact 7 (46.7%) had > 118.8 and 8 (53.3%) had <118.8 ng/ml periostin level. If periostin is not significantly different in PTB and non-PTB groups, what was the purpose to include it is classification tree and how is it helpful to discriminate the patient groups?

As we mentioned in the Materials and Methods section (lines 132-135), four proteins, MMP-1, MMP-9, IFN-gamma, and periostin were included in the classification tree analysis, although IFN-gamma was not selected as a marker for PTB diagnosis. The classification method automatically chooses appropriate markers and their cut-off values to divide the patients into two groups (in this study, PTB and non-PTB) according to its algorithm, which is totally different from ROC. The rationale of the classification tree in the study was also explained in the Abstract (lines 36-37) and Discussion (lines 286-289) sections.

We really appreciate your comments regarding the description in line 211 in the previous version. The explanation of box C in Figure 4 was incorrect in the previous version. We corrected this sentence (line 215) in the revised manuscript.

- Similarly, Infiltration>5 group, nearly equal numbers of PTB patients have MMP-9 <3.009 (N=14) and >3.009 (N=15).What is the relevance of these values.

Thank you for your comments. The important point is that non-PTB patients were not included in Box D in Figure 4.

- I highly recommend along with Figure 5 (This is actually a Table) authors include a summary diagram (diagnostic algorithm) that is easy to follow and explains criteria / variables / cut-off values proposed by the authors to discriminate PTB and non-PTB groups.

Thank you for your comments. We already explained the interpretation of the classification tree results in the Discussion section (lines 292-296). If a pneumonia patient with limited infiltration is dropped into “Box C” in Figure 4, the patient is likely not to have PTB (95%CI 0.0-41.0). On the other hand, if a pneumonia patient with extensive infiltration is dropped into “Box D” in Figure 4, the patient is likely to have PTB (95%CI 76.8-100.0).

Reviewer #2: In this study, the authors analyzed the serum levels of IFN-γ, MMP-1, MMP-9 and periostin in tuberculous pneumonia and non-tuberculous pneumonia patients. They also developed a new classification tree by using MMP-1, MMP-9 and periostin level to diagnose tuberculous pneumonia and non-tuberculous pneumonia patients. Their results are of interest. However, there are several major concerns that need to be addressed.

We really appreciate your reading of our manuscript and your comments and suggestions. We revised our manuscript following your comments. It would be most appreciated if you could give us your feedback to our response.

1. MMP-1 and MMP-9 are well known factors involved in many diseases. They are not specific markers for tuberculosis. The sample size in this study is small (40 vs. 28) and patients in each group are with a wide range of ages and diseases. I think it will be more appropriate to perform a more specific analysis. E.g. compare PTD vs non-PTD when both group patients have cardiovascular disorder or bilateral disease (with similar condition)…

Thank you for your comments. As you suggested, we understand that MMP-1 and MMP-9 are not specific markers for PTB, and several conditions might affect these levels in the serum. Following your suggestion, we compared serum IFN-� and MMP-1 levels (these two were evaluated because these were significantly higher in PTB compared to non-PTB) by gender and underlying diseases. No significant differences were observed in serum IFN-γ levels by gender or underlying diseases. In contrast, serum MMP-1 levels in patients with malignancy were significantly higher than in those without malignancy (Figure S1). Therefore, we performed multivariate analysis and showed that only PTB was associated with elevated serum MMP-1 levels (Table S2). We think the more specific analysis you suggested is not appropriate in this case because the number of patients becomes smaller and the statistic power weaker.

2. The authors need to clarify the expression profile of MMP-1, MMP-9 and periostin over the infection time. Is there any dynamics? I am not clear if they compared the samples from the same time after infection.

Thank you for your comments. We understand that the duration between the time of infection (or onset of symptoms) and serum collection is important, although it is difficult to determine the time of infection (or onset of symptoms) because of the nature of TB. In terms of kinetics of these proteins, we compared serum periostin levels between pre- (N=16) and post- (N=16) TB treatment, although no significant difference was observed. We totally agree that the data regarding kinetics of these biomarkers deepen our understanding. However, this was not the primary purpose of the study, and serum samples collected after TB treatment were stored in only a subset of patients. This issue will be clarified in future studies.

3. I hope they can provide more explanation on the association between MMP-9 level and pneumonia patients with limited infiltration; also for the association between MMP-1, periostin and pneumonia patients with extensive infiltration.

Thank you for your comments. We assume this comment is related to Figure 4. The classification tree method automatically chooses appropriate markers and their cut-off values to divide patients into two groups (in this study, PTB and non-PTB) according to its algorithm. Therefore, the interpretation is sometimes difficult. However, we added our interpretation of the results in Figure 4 to the Discussion section (lines 298-305).

4. I feel a little bit hard to read this manuscript due to the language. I suggest they have someone to correct the grammar and typos.

We asked Editage (www.editage.com) for English language editing. We added this information to the Acknowledgements section.

---

## [Decision Letter · Decision Letter 1]

26 Dec 2019

Diagnostic performance of serum interferon gamma, matrix metalloproteinases, and periostin measurements for pulmonary tuberculosis in Japanese patients with pneumonia

PONE-D-19-19256R1

Dear Dr. Kinjo,

We are pleased to inform you that your manuscript has been judged scientifically suitable for publication and will be formally accepted for publication once it complies with all outstanding technical requirements.

With kind regards,

Frederick Quinn

Academic Editor

PLOS ONE

Additional Editor Comments (optional):

Reviewers' comments:

Reviewer's Responses to Questions

**Comments to the Author**

1. If the authors have adequately addressed your comments raised in a previous round of review and you feel that this manuscript is now acceptable for publication, you may indicate that here to bypass the “Comments to the Author” section, enter your conflict of interest statement in the “Confidential to Editor” section, and submit your "Accept" recommendation.

Reviewer #1: All comments have been addressed

Reviewer #2: All comments have been addressed

2. Is the manuscript technically sound, and do the data support the conclusions?

Reviewer #1: Partly

Reviewer #2: Yes

3. Has the statistical analysis been performed appropriately and rigorously? 

Reviewer #1: Yes

Reviewer #2: Yes

4. Have the authors made all data underlying the findings in their manuscript fully available?

Reviewer #1: Yes

Reviewer #2: Yes

5. Is the manuscript presented in an intelligible fashion and written in standard English?

Reviewer #1: Yes

Reviewer #2: Yes

6. Review Comments to the Author

Reviewer #1: (No Response)

Reviewer #2: (No Response)

7. PLOS authors have the option to publish the peer review history of their article (what does this mean?). If published, this will include your full peer review and any attached files.

Reviewer #1: No

Reviewer #2: No

---

## [Editor Report · Acceptance letter]

30 Dec 2019

PONE-D-19-19256R1 

Diagnostic performance of serum interferon gamma, matrix metalloproteinases, and periostin measurements for pulmonary tuberculosis in Japanese patients with pneumonia 

Dear Dr. Kinjo:

I am pleased to inform you that your manuscript has been deemed suitable for publication in PLOS ONE. Congratulations! Your manuscript is now with our production department. 

With kind regards,

on behalf of

Dr. Frederick Quinn 

Academic Editor

PLOS ONE